# Polyphenol-Rich Extract from ‘Limoncella’ Apple Variety Ameliorates Dinitrobenzene Sulfonic Acid-Induced Colitis and Linked Liver Damage

**DOI:** 10.3390/ijms25063210

**Published:** 2024-03-11

**Authors:** Stefania Lama, Ester Pagano, Francesca Borrelli, Maria Maisto, Gian Carlo Tenore, Maria Francesca Nanì, Pilar Chacon-Millan, Ettore Novellino, Paola Stiuso

**Affiliations:** 1Department of Precision Medicine, University of Campania “Luigi Vanvitelli”, Via de Crecchio 7, 80138 Naples, Italy; stefania.lama@unicampania.it (S.L.); pilar.chaconmillan@unicampania.it (P.C.-M.); 2Department of Pharmacy, University of Naples Federico II, 80131 Naples, Italy; ester.pagano@unina.it (E.P.); francesca.borrelli@unina.it (F.B.); maria.maisto@unina.it (M.M.); giancarlo.tenore@unina.it (G.C.T.); mariafrncesca.nani@unina.it (M.F.N.); 3Department of Medicine and Surgery, Università Cattolica del Sacro Cuore, 00168 Rome, Italy; ettore.novellino@unicatt.it

**Keywords:** nutraceuticals, apple polyphenols, inflammatory bowel disease, antioxidant, liver inflammation

## Abstract

Inflammatory bowel conditions can involve nearly all organ systems and induce pathological processes through increased oxidative stress, lipid peroxidation and disruption of the immune response. Patients with inflammatory bowel disease (IBD) are at high risk of having extra-intestinal manifestations, for example, in the hepatobiliary system. In 30% of patients with IBD, the blood values of liver enzymes, such as AST and ALT, are increased. Moreover, treatments for inflammatory bowel diseases may cause liver toxicity. Apple polyphenol extracts are widely acknowledged for their potential antioxidant effects, which help prevent damage from oxidative stress, reduce inflammation, provide protection to the liver, and enhance lipid metabolism. The aim of this study was to investigate whether the polyphenol apple extract from Malus domestica cv. ‘Limoncella’ (LAPE) may be an effective intervention for the treatment of IBD-induced hepatotoxicity. The LAPE was administrated in vivo by oral gavage (3–300 mg/kg) once a day for 3 consecutive days, starting 24 h after the induction of dinitro-benzenesulfonic acid (DNBS) colitis in mice. The results showed that LAPE significantly attenuated histological bowel injury, myeloperoxidase activity, tumor necrosis factor and interleukin (IL-1β) expressions. Furthermore, LAPE significantly improved the serum lipid peroxidation and liver injury in DNBS-induced colitis, as well as reduced the nuclear transcription factor-kappaB activation. In conclusion, these results suggest that LAPE, through its antioxidant and anti-inflammatory properties, could prevent liver damage induced by inflammatory bowel disease.

## 1. Introduction

Inflammatory bowel diseases (IBDs), such as Crohn’s disease (CD) and ulcerative colitis (UC), involve persistent inflammation in the digestive tract. Notably, more than 35% of IBD patients are affected by extraintestinal manifestations (EIMs), which significantly impact the quality of life and require additional treatments. The EIMs include arthropathies, mucocutaneous and ophthalmological manifestations, as well as conditions affecting the hepatobiliary system. Among these, primary sclerosing cholangitis (PSC) is the most common extraintestinal manifestation in IBD, affecting 1.4–7.5% of patients. The gastrointestinal tract is anatomically connected to the hepatobiliary system via the mesenteric and portal veins. Thus, the liver and the biliary system are more susceptible to colonic inflammatory response. Moreover, although the pathogenesis is multifactorial, IBD is associated with an increased intestinal permeability, which provides a favorable environment for the translocation of gut flora to the liver. Further, during recurrent episodes of active colitis, the hepatic microcirculation recruits leukocytes to the sinusoids [1,2,3], leading to liver inflammation and hepatocellular damage.

Polyphenols, a large group of phytochemicals, have been shown to be effective in the prevention and treatment of several diseases [4], mainly due to their antioxidant, anti-inflammatory and antihyperlipidemic activities. The anti-inflammatory effect of polyphenols is mainly due to the modulation of the nuclear factor kappa-light-chain-enhancer of activated B cells (NF-kB) and the reduction of pro-inflammatory cytokines, which represent key targets for the immune therapy in IBD patients [5]. In the last decade, several studies have reported the contribution of polyphenols, such as chlorogenic acid, isoflavones [6], proanthocyanidin [7,8,9] and catechins [10], in maintaining intestinal barrier functions through their antioxidant effects [11,12,13] and in alleviating mucosal inflammation by inhibiting the canonical NF-kB pathway. Moreover, the polyphenols curcumin, quercetin, epigallocathechin-3-gallate and resveratrol have been reported to inhibit the wingless-related integration site (WNT)-activated pathways [13]. In the intestinal epithelium, the Wnt-signaling pathway promotes proliferation and/or differentiation of stem cells in the intestinal crypts, the epithelial mesenchymal transition (EMT) and the aberrant expansion of cancer stem cells (CSC).

Malus domestica cv. ‘Limoncella’ is a juicy and aromatic apple cultivar [13] rich in nutritional polyphenols, such as phenolic acids, anthocyanins, flavonoids, and proanthocyanidins, which have several health-promoting effects, including antioxidant and anti-inflammatory activities [14]. The Limoncella apple develops a yellow peel with a citric taste that lasts until the end of spring. Recently, Riccio et al. have demonstrated the inhibitory effect of Limoncella apple polyphenol extracts (LAPE) on the Wnt/β-catenin pathway in tumor colonic epithelial cells [15]. 

In this study, we have evaluated the therapeutic potential of LAPE in a model of colitis associated with liver damage. Hence, the pharmacological effect of LAPE was investigated in a mouse model of colitis induced by 2,4,6-dinitrobenzenesulfonic acid (DNBS) [16]. Colonic inflammation was studied by monitoring macroscopic parameters, histological damage, cytokine production and intestinal permeability. Therefore, the potential effect of LAPE on colitis-induced liver damage was evaluated by assessing biochemical parameters, inflammatory markers and lipid accumulation. Here, we propose LAPE for the treatment of intestinal inflammation and prevention of colitis-induced liver injury.

## 2. Results

### 2.1. LAPE Extract Composition

The polyphenolic content of LAPE was evaluated by HPLC-DAD analysis (the results were reported in Appendix A) and expressed as mg/g of total extract: gallic acid 0.133 ± 0.004 (2.5%); procyanidin B1 0.09 ± 0.007 (1.9%); catechin 0.86 ± 0.005 (16.5%); chlorogenic acid 1.54 ± 0.023 (29.6%); procyanidin B2 0.278 ± 0.003 (0.53%); epicatechin 0.505 ± 0.002 (9.6%); procyanidin C1 0.21 ± 0.06 (4%); rutin 0.038 ± 0.001 (0.73%); phlorizin 1.60 ± 0.021 (30.7%). The total amount of identified polyphenols in 1 g of LAPE was 5.25 mg; thus, according to our experimental model, each mouse received 47.5 mg of titrated polyphenols in 9 mg of LAPE administrated daily.

### 2.2. LAPE Ameliorates the Macroscopic and Histopathological Changes in the DNBS Model of Colitis

The flogogen agent DNBS caused several signs of intestinal inflammation, including increased colon weight/colon length ratio, histological damage, neutrophil infiltration and intestinal permeability (Figure 1, Figure 2 and Figure 3). Oral administration of LAPE reduced the colon weight/colon length ratio increased by DNBS, with a significant effect starting from the 30 mg/kg dose (Figure 1A). Moreover, LAPE, at a dose of 300 mg/kg, was also able to reduce the DNBS-induced weight loss at day 4 (% of mice weight loss ± SEM: control: 113.33 ± 3.0 ****; DNBS: 81.9 ± 1.8; DNBS+LAPE: 94.2 ± 4.0 *; * *p* < 0.05 and **** *p* < 0.0001 vs. DNBS). Further, the histological evaluation (Figure 1B) of the colonic specimens of DNBS-treated mice confirms the anti-inflammatory activity of LAPE. The H&E-stained colon of NBS-treated mice was characterized by cellular infiltration, edemas and the presence of inflammatory lesions (Figure 1B), compared to control mice. As shown in Figure 1B, LAPE administration decreased the histological signs of colon injury, such as crypt loss and the inflammatory cell infiltrate, compared to mice treated with DNBS.

### 2.3. LAPE Restores Colonic Neutrophil Infiltration and Cytokine Production in DNBS-Treated Mice 

The anti-inflammatory effect of LAPE (at the dose of 300 mg/kg) on DNBS-induced colitis was confirmed by further analyses, using the more effective dose (i.e., 300 mg/kg). Neutrophil infiltration was biochemically measured by assessing MPO activity in the inflamed colonic specimens. MPO is a peroxidase that reflects neutrophil infiltration and colonic inflammation. DNBS increased the colonic MPO, which was significantly reduced by oral gavage (300 mg/kg) administration of LAPE (Figure 2A). Moreover, we evaluated the main pro-inflammatory cytokines (i.e., IL-1β and IL-6) involved in colitis. In comparison with the control group, DNBS-treated mice showed a significant increase in colonic production of IL-1β (Figure 2B) and IL-6 (Figure 2C). LAPE administration (300 mg/kg, by oral gavage) significantly reduced IL-1β and IL-6 colonic levels compared to the DNBS group. 

### 2.4. LAPE Restores the Intestinal Permeability in the DNBS Model of Colitis 

It is widely reported that experimental and clinical colitis are associated with an increased intestinal permeability. Accordingly, DNBS significantly increased intestinal permeability compared to control mice, as shown by the higher serum concentration of FITC-dextran after its oral administration (Figure 3). As shown in Figure 3, LAPE (300 mg/kg, by oral gavage) significantly reduced the increase in FITC-dextran serum levels, which is used as an indirect marker of intestinal permeability (Figure 3). 

Since β-catenin promotes cell adhesion and, importantly, the Wnt/β-catenin pathway is involved in the modulation of intestinal permeability, we also investigated the effect of LAPE on the colonic expression of β-catenin. Immunofluorescence analysis showed that DNBS caused a significant decrease in the levels of β-catenin on the cell membrane, while LAPE treatment induced the β-catenin relocalization at membranous level in DNBS-treated mice (Figure 4), promoting cell–cell adhesion and preventing the intestinal barrier disruption.

### 2.5. LAPE Reduces Liver Dysfunction and Hepatic Lipidic Accumulation in the DNBS Model of Colitis 

The increase in colon permeability leads to the translocation of gut microbiota to liver and to liver inflammation and damage. In our experiments, DNBS caused a significant increase in serum liver-specific enzymes (i.e., ALT and AST) and oxidative stress, evaluated by lipid peroxidation (Table 1). In DNBS-treated mice, oral administration of LAPE resulted in an approximately two-fold reduction in the enzymes ALT and AST compared to DNBS-induced colitis mice (Table 1). Furthermore, LAPE significantly reduced lipid peroxidation by three-fold in DNBS-treated mice, as assessed by the TBARS assay (Table 1). 

Furthermore, liver sections from DNBS-treated mice were stained with Oil Red O, a fat-soluble dye that marks neutral triglycerides (TG) and lipids. Compared to control mice, DNBS caused an increase in lipid accumulation in the mice liver, which was counteracted by LAPE administration (Figure 5A). In line with these results, LAPE reduced the increased levels of TG in the liver caused by DNBS (Figure 5B). 

Moreover, histopathological examination of liver sections observed by H&E staining in DNBS administered mice (Figure 6), showed both inflammatory cells, infiltration and eroded portal triad compared to the control mice. The LAPE treatment of DNBS-induced colitis mice showed the inflammation liver reduction and a recovery of the damage of hepatocytes recovered compared to DNBS treated mice. Liver H&E staining revealed no apparent vacuolar degeneration in DNBS-treated mice compared to controls. Furthermore, the nuclear transcription factor-kB (NF-kB), evidenced by immunofluorescence, (Figure 7) was increased in DNBS treated mice compared with control mice, while the NF-kB immunofluorescence in LAPE-treated DNBS-mice liver tissue was comparable to control mice.

## 3. Discussion

IBDs, such as ulcerative colitis and Crohn’s disease, are chronic relapsing and remitting inflammatory conditions of the gastrointestinal tract. During IBD, pro-inflammatory mediators (such as cytokines) activate the NF-kB and induce an increase in reactive oxygen species (ROS) [18,19]. Therefore, the control of inflammation and oxidative stress plays an important role in the treatment of IBD. Notably, chronic intestinal inflammation, as well as the adverse reactions of corticosteroid therapy, are often associated with liver damage [20] and can lead to chronic liver disease [21]. Therefore, it is crucial to find new compounds without side effects, which, by reducing intestinal inflammation, could also decrease or prevent liver damage. 

Several in vitro and in vivo studies have revealed the preventive and therapeutic effects of polyphenols on intestinal inflammation [22,23]. Recently, it has been shown that LAPE inhibits the canonical Wnt pathway in the colonic epithelial cells [15]. Notably, Wnt/β-catenin signaling is a key regulator of intestinal epithelial homeostasis [24]. 

In this study, we demonstrated that a polyphenol extract of Limoncella apple with a high content of chlorogenic acid, catechin and epicatechin ameliorates the DNBS-induced intestinal and liver inflammation. In mice treated with DNBS, we have shown that oral administration of LAPE reduced the histological damage in the inflamed colon, as well as the colon weight/colon length ratio, a macroscopic marker of colitis severity [16]. The intestinal anti-inflammatory effect of LAPE was further confirmed by its ability to reduce the production of the pro-inflammatory cytokines IL-1β and IL-6 in colon tissues. It is well-established that pro-inflammatory mediators induce neutrophil infiltration, leading to the amplification of the inflammatory response and intestinal injury. Accordingly, LAPE was also able to reduce MPO activity, a known marker of neutrophil infiltration in colitis [16]. 

As the dysfunction of the intestinal barrier is a crucial factor in the development of IBD, we assessed the impact of LAPE on the integrity of the intestinal epithelium. In the present study, we demonstrated that oral administration of LAPE restored the DNBS-increased intestinal permeability, as well as increased the β-catenin localization at the colon membrane, maintaining cell–cell adhesion and preventing barrier disruption. β-catenin is a key regulator of intestinal epithelial renewal and the main nuclear effector of canonical Wnt signaling [13,24,25]. Moreover, the inhibition of β-catenin translocation to nucleus after LAPE treatment suggests inactivation of the Wnt pathway, confirming the results previously reported in colonic epithelial cells [15].

It is well established that hepatobiliary pathologies are the most common extraintestinal manifestations in IBD patients [26]. Relevant to this point, LAPE administration reduced the histological damage to the liver caused by colitis as well as the accumulation of the lipid in the liver tissue. Furthermore, LAPE ameliorated the hepatic functionality in mice with colitis, as revealed by the reduction in the levels of ALT and AST. Since the Wnt/β-catenin signaling pathway modulates inflammatory and immune responses via NF-κB [26], we evaluated the effect of LAPE on the expression of this nuclear factor in liver tissues. The NF-κB pathway requires a delicate balance, as too little or too much NF-κB activation may have negative effects on the liver, either by exacerbating liver inflammation or failing to adequately protect it from cell death [27]. Our study demonstrates that LAPE exerts an anti-inflammatory effect on liver injury by inhibiting NF-κB activation at the concentration of 300 mg/kg. Specifically, this treatment dose for animals is equivalent to a human daily dose of 40.50 g of Limoncella apple (fresh weight). Since this peculiar apple variety produces small and elongated fruits with a reported average weight of 100 g, the theoretically calculated daily dose for humans could be equated to consuming approximately half of a single Limoncella apple per day.

## 4. Materials and Methods

### 4.1. Drugs and Reagents

2,4,6-dinitrobenzenesulfonic acid (DNBS), myeloperoxidase (MPO), fluorescein isothiocyanate (FITC)-conjugated dextran (molecular mass 3–5 kDa), 4′,6-Diamidino-2-phenylindole (DAPI) and bovine serum albumin (BSA) were acquired from Sigma (Milan, Italy). Phosphate-buffered saline (PBS) was purchased from Lonza (Milan, Italy). All buffers and solutions were prepared with ultra-high-quality water. All reagents were of the purest commercial grade. All chemicals and reagents employed in this study were of analytical grade.

### 4.2. Limoncella Apple Polyphenol Extract (LAPE) Preparation 

Apples of Malus domestica B. cv. ‘Limoncella’ were gathered in October, after harvesting, in Castelvetere del Calore (Avellino, Italy, Coordinates: 40°55′47″ N 14°59′13″ E). To prepare Limoncella apple extract, lyophilized peels and flesh (10 g) from Limoncella apple samples underwent treatment with 60 mL of 80% methanol (0.5% formic acid). The mixture was homogenized for 5 min using an ultra-turrax (T25-digital, IKA, Staufen im Breisgau, Berlin, Germany) and agitated on an orbital shaker (Sko-DXL, Argolab, Carpy, Italy) at 300 rpm for 15 min. Subsequently, the samples were subjected to an ultrasonic bath treatment for an additional 10 min before being centrifuged at 4427× *g* for 10 min. The resulting supernatants were collected and stored in darkness at 4 °C. The obtained pellets underwent a second extraction process following the same procedure with an additional 40 mL of the solvent mixture. Finally, the extracts were vacuum filtered, the methanol fraction was removed through evaporation, and the water fraction was lyophilized. The extracts were stored at −20 °C until analysis.

### 4.3. HPLC/DAD Analysis

Analyses of Limoncella apple peel extract (LAPE) were conducted using a Jasco ExtremaLC-4000 system (Jasco Inc., Easton, MD, USA) equipped with a photodiode array detector (DAD). The chosen column was a Kinetex^®^ C18 column (250 mm × 4.6 mm, 5 μm; Phenomenex, Torrance, CA, USA). The analyses were carried out at a flow rate of 1 mL/min, using solvent A (2% formic acid) and solvent B (0.5% formic acid in acetonitrile and water 50:50, *v*/*v*). After a 5 min hold at 10% solvent B, elution proceeded with the following conditions: a gradient from 10% (B) to 55% (B) over 50 min, followed by an increase to 95% (B) in 10 min, with a 5 min maintenance period. Flavanols, procyanidins, dihydrochalcones and hydroxycinnamic acids were acquired at 280 nm, while flavan-3-ols were recorded at 360 nm. For quantitative determination, standard curves were established for each polyphenol standard across a concentration range of 0.1–1.0 μg/μL, with six different concentration levels and duplicate injections at each level. The identity of polyphenols was confirmed by the comparison of the retention time with analytical standards from Sigma-Aldrich Chemical Co. (St. Louis, MO, USA).

### 4.4. Animals

Male CD1 mice (25–30 g, 6 to 8 weeks old) were supplied from Charles River Laboratories (Calco, Lecco, Italy) and housed at the animal house of the Department of Pharmacy (University of Naples Federico II) in polycarbonate cages under controlled temperature (23 ± 2 °C), constant humidity (60%) and with a 12 h light and 12 h dark cycle. Mice were acclimatized for 1 week under standard environment conditions, with free access to tap water, and fed ad libitum on a standard rodent diet. All mice were fasted overnight before the intracolonic injection of DNBS and for 2 h before the oral gavage of LAPE. All experimental procedures and protocols were approved by the Institutional Animal Ethics Committee and the Italian Ministry of Health (protocol number 444/2018) and were carried out in accordance with the Italian D.L. no. 116 of 27 January 1992 and associated guidelines in the European Communities Council (86/609/ECC and 2010/63/UE). All studies involving mice are reported in compliance with the ARRIVE guidelines for the use of experimental animals [16].

### 4.5. DNBS-Induced Experimental Colitis and Pharmacological Treatments

Colitis was induced by intrarectal administration of DNBS (150 mg/kg) by a polyethylene catheter (1 mm in diameter) inserted approximately 4.5 cm proximal to the anus. DNBS was dissolved in 50% (*v*/*v*) ethanol/water solution (150 μL/mouse). The DNBS dose was selected based on our preliminary experiments, showing remarkable colonic damage associated with high reproducibility and low mortality. All mice were euthanized by asphyxiation with CO_2_ 3 days after the administration of DNBS, when intestinal inflammation can be easily assessed. Then, the mice’s abdomens were opened by a midline incision, and the colons were removed, isolated from surrounding tissues, their length measured, rinsed, weighed, and processed as required. Mice were randomly allocated to 5 experimental groups as follows: control (vehicle), DNBS, DNBS+LAPE 3 mg/kg, DNBS+LAPE 30 mg/kg and DNBS+LAPE 300 mg/kg (10–12 mice for each experimental group). Outcome assessments were performed in a single-blind manner. The vehicle (water) or LAPE were given by oral gavage (3–300 mg/kg) once a day for 3 consecutive days starting 24 h after the colitis induction. The last administration of vehicle or LAPE was given 2 h before euthanasia. The colon weight/colon length ratio (mg/cm) was used as a macroscopic marker of intestinal inflammation. For biochemistry analysis, tissues were snap-frozen and kept at −80 °C until use, while for histopathological analysis, samples were fixed in 10% formaldehyde. 

### 4.6. In Vivo Intestinal Permeability

Intestinal permeability was evaluated using the FITC-labeled dextran method in mice, involving blood collection through cardiac puncture [19]. In brief, two days after administering DNBS, mice received 600 mg/kg body weight of fluorescein isothiocyanate (FITC)-conjugated dextran (molecular mass 3–5 kDa) via oral gavage. One day later, mice were euthanized using CO_2_ asphyxiation, blood was collected through cardiac puncture and the serum was analyzed for FITC-derived fluorescence using a fluorescent microplate reader with excitation–emission wavelengths of 485–520 nm (LS55 Luminescence Spectrometer, PerkinElmer Instruments, Shanghai, China). A standard curve was generated using serial dilutions of FITC-dextran in tap water, and then the fluorescein fluorescence in samples and standards was measured. Intestinal permeability was quantified as FITC (μM) detected in the mice serum.

### 4.7. Measurements of Myeloperoxidase (MPO) Activity

MPO activity, a peroxidase enzyme used to measure the neutrophil infiltration in inflamed colons, was assessed as previously described. Briefly, full-thickness colons were mechanically homogenized in a lysis buffer (0.5% hexadecyltrimethylammonium bromide in 3-(N morpholino) propane sulfonic acid 10 mM) in the ratio of 50 mg tissue/mL buffer. The homogenates were centrifuged for 20 min at 15,000× *g* at 4 °C and the supernatant was incubated with a sodium phosphate buffer (NaPP pH 5.5) and tetra-methylbenzidine 16 mM. After 5 min, 1 M hydrogen peroxide was added, then the reaction was stopped with 2 M acetic acid. The rate of MPO activity was measured by using a spectrophotometer cuvette (650 nm). Different dilutions of human MPO enzyme of known concentrations were used to obtain a standard curve. MPO activity was calculated using standard curve interpolation and expressed as U/mg of colon tissue.

### 4.8. Measurement of Colonic Cytokine Production 

Interleukin (IL)-1β, IL-6 and IL-10 levels were measured in the colon homogenate supernatants obtained from all experimental groups by using commercial enzyme-linked immunosorbent assay (ELISA) kits (Invitrogen, Waltham, MA, USA) according to the manufacturer’s instructions. Briefly, full-thickness colons were homogenized in a lysis buffer containing 0.5 M b-glycerophosphate, 20 mM MgCl_2_, 10 mM ethylene glycol tetraacetic acid and supplemented with 100 mM dithiothreitol and protease/phosphatase inhibitors (100 mM dimethylsulphonyl fluoride, 2 mg/mL apronitin, 2 mM leupeptin and 10 mM Na_3_VO_4_). Homogenates were centrifuged at 600× *g* for 10 min at 4 °C, then the supernatants were collected, centrifuged at 15,000× *g* for 10 min at 4 °C and used for cytokine detection. The absorbance was measured at 490 nm and the results were expressed as pg/mL of protein extract.

### 4.9. Histological Analysis

Paraffin-embedded colon and liver tissues from mice were cut into 5 mm sections. The sections were stained with hematoxylin-eosin (H&E), according to the protocol previously described [21]. After dehydration, the sections were examined under a light microscope (Leica DM 2500, Leica Microsystems, Wetzlar, Germany). Photographs were taken using a Leica DFC320 R2 digital camera. 

### 4.10. Serum ALT and AST and Lipid Peroxidation Analysis

To measure the biochemical markers of hepatic injury, the serum levels of alanine transaminase (ALT) and aspartate transaminase (AST) were analyzed by colorimetric methods. All measurements were performed using commercially available diagnostic kits for AST/ALT activity (cat. N. MAK055/MAK052, Sigma-Aldrich, St. Louis, MO, USA). Further, the serum lipid peroxidation was evaluated by the thiobarbituric acid reactive substance (TBARS) assay. Briefly, sera samples were incubated with 0.5 mL of aqueous solution of thiobarbituric acid and acetic acid at pH 3.5. After heating at 95 °C for 45 min, the samples were centrifuged at 4000 rpm for 5 min. In the supernatant fractions, TBARS were quantified by spectrophotometry at 532 nm. Results were expressed as TBARS μM/μg of serum protein. Each data point is the average of triplicate measurements, with each individual experiment performed in duplicate.

### 4.11. Immuno-Fluorescence Analysis

The fixed samples (colon and liver) were dewaxed, re-hydrated and processed. Briefly, antigen retrieval was carried out by pressure-cooking slides for 3 min in a 0.01 M citrate buffer (pH 6.0). To avoid nonspecific interactions of antibodies, the slides were treated for 3 h in 5% BSA in PBS. Immunostaining of colon tissue was performed by overnight incubation at 4 °C, with an anti-β-catenin antibody (1:500, Alexa Fluor^®^, BD Pharmingen^TM^, Franklin Lakes, NJ, USA). Immunostaining of liver tissue was performed by overnight incubation at 4 °C with an anti-p50 NF-kB antibody (1:100, Santa Cruz Biotechnology, Dallas, TX, USA). After washing in PBS, the slides were incubated for 1 h with a secondary 633-conjugated antibody (1:500 Alexa Fluor, Invitrogen). The slides were mounted on microscope slides using Mowiol + DAPI for nuclear staining and then observed under the optical microscope (Leica DM 5000 B + CTR 5000) [17].

### 4.12. Statistical Analysis

All results were expressed as means ± standard error of the mean (SEM) of at least three independent experiments. Statistical analysis was performed using GraphPad Prism version 8. All data were compared using a one-way analysis of variance (ANOVA), followed by Dunnett’s multiple comparison test. The symbol “n” refers to the number of samples for each set of experiments. Statistically significant differences were accepted when *p* was <0.05. 

## 5. Conclusions

In this study, we demonstrated that LAPE ameliorated intestinal inflammation in mice treated with the flogogen agent DNBS. In addition, LAPE decreased serum lipid peroxidation and liver inflammation by inhibiting NF-kB in the experimental model of colitis. However, additional experiments are needed to further validate the results obtained in our mouse model exploratory study and elucidate the impact of LAPE as a protective dietary supplement in chronic intestinal inflammation leading to liver injury. 

## Figures and Tables

**Figure 1 ijms-25-03210-f001:**
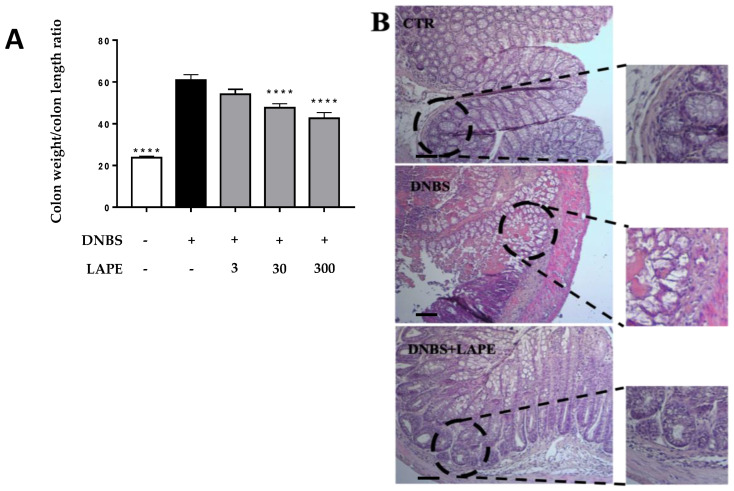
Effect of LAPE on colonic damage in DNBS-treated mice. Effect of LAPE (30–300 mg/kg, given by oral gavage) on colon weight/colon length ratio (**A**) and, at the dose of 300 mg/kg, on colonic damage (**B**) in DNBS-induced colitis. All acquisitions were performed 3 days after DNBS injection. LAPE was given once a day starting from 24 h after DNBS and continued for the following days until the sacrifice. (**A**) Data are expressed as the mean ± SEM of 10–12 mice for each experimental group and were statistically analyzed using a one-way ANOVA followed by Dunnett’s test. **** *p* < 0.0001 versus DNBS alone. (**B**) Representative H&E-stained colon cross-sections of mice treated with vehicle, DNBS and DNBS+LAPE (300 mg/kg by oral gavage). Original magnification 10×. Scale bars represented 100 μm.

**Figure 2 ijms-25-03210-f002:**
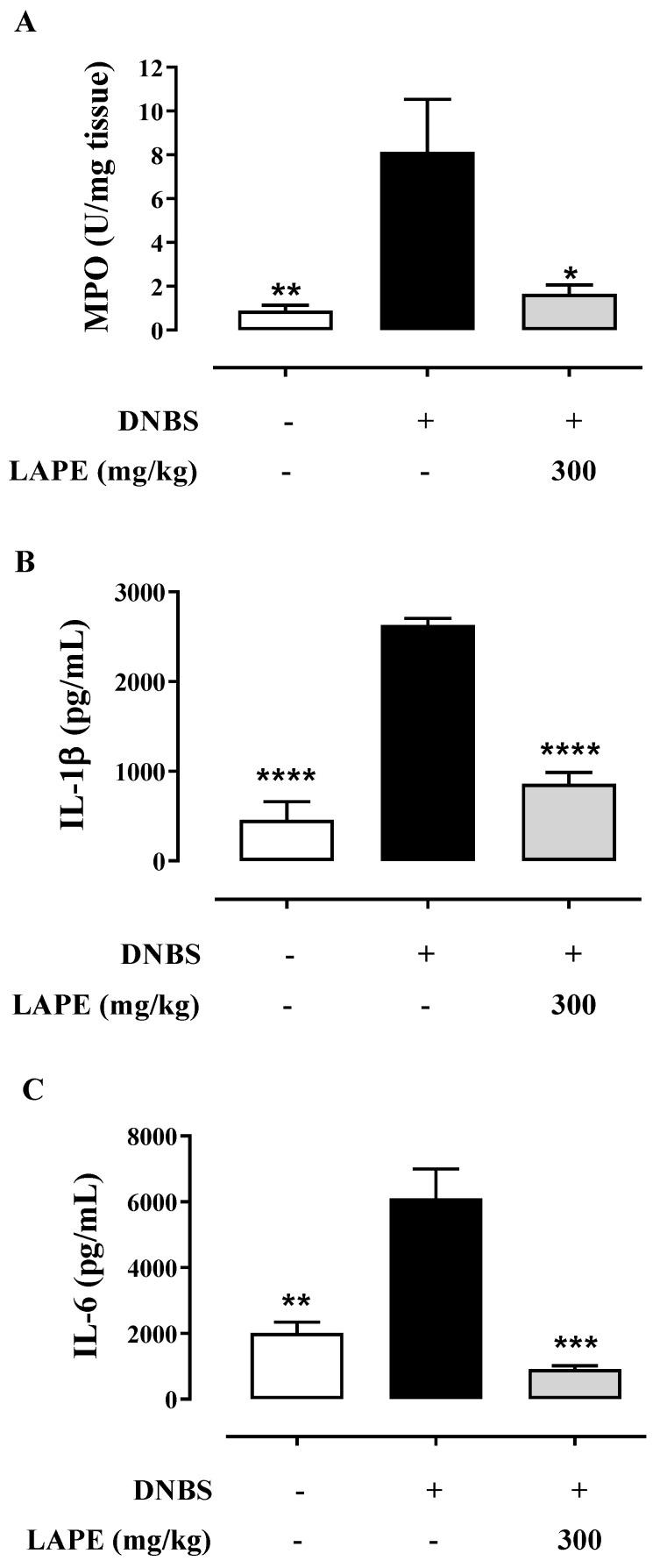
Effect of LAPE on inflammatory parameters in DNBS-treated mice. Effect of LAPE (300 mg/kg, given by oral gavage) on myeloperoxidase (MPO) activity (**A**) and on colonic IL-1β (**B**) and IL-6 (**C**) levels in DNBS-induced colitis. Measurements were performed 3 days after DNBS administration. LAPE was given once a day starting from 24 h after DNBS and continued for the following days until the sacrifice. MPO, IL-1β and IL-6 levels were measured in vehicle-, DNBS- or DNBS+ LAPE-treated mice. Data are expressed as the mean ± SEM of 5 (**A**) or 4 (**B**,**C**) mice for each experimental group and were statistically analyzed using a one-way ANOVA followed by Dunnett’s test. * *p* < 0.05, ** *p* < 0.01, *** *p* < 0.001 and **** *p* < 0.0001 vs. DNBS alone.

**Figure 3 ijms-25-03210-f003:**
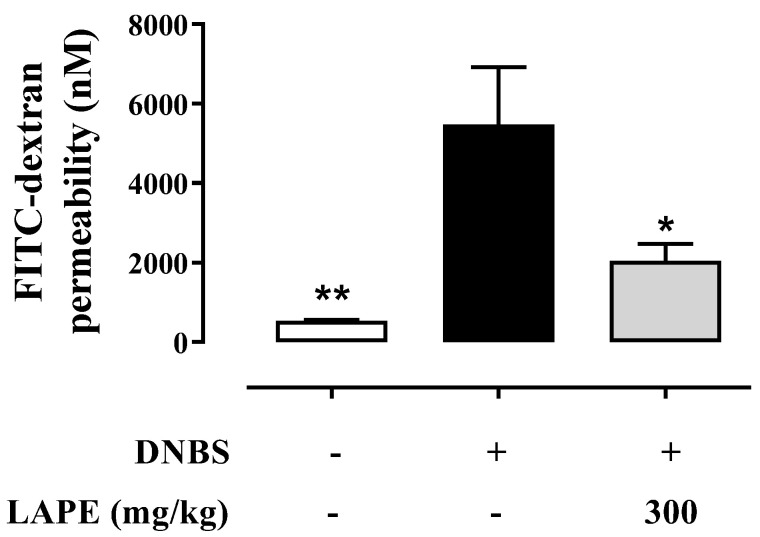
Effect of LAPE on intestinal permeability in DNBS-treated mice. Effect of LAPE (300 mg/kg, given by oral gavage) on serum FITC–dextran concentration in DNBS-induced colitis. Measurements were performed 3 days after DNBS injection. LAPE was given once a day starting from 24 h after DNBS and continued for the following days until the sacrifice. Data are expressed as the mean ± SEM of 4 mice for each experimental group and were statistically analyzed using a one-way ANOVA followed by Dunnett’s test. * *p* < 0.05 and ** *p* < 0.01 vs. DNBS alone.

**Figure 4 ijms-25-03210-f004:**
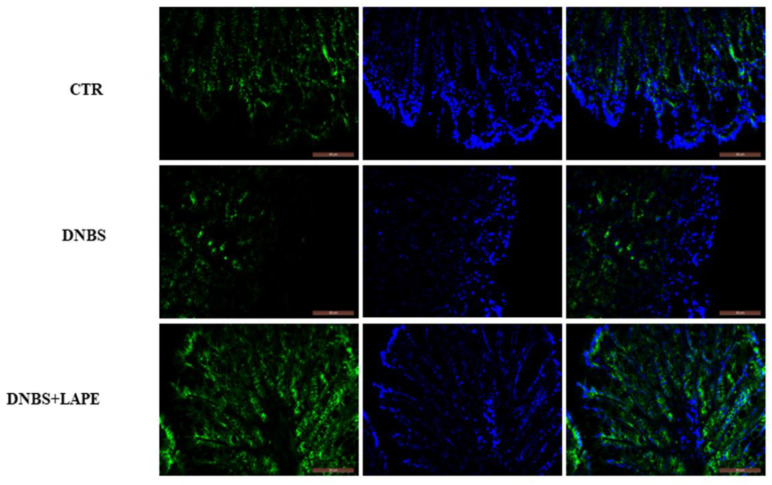
Effect of LAPE on β-catenin localization in DNBS-treated mice. Representative image of colon cross-sections showing the expression of β-catenin (green) and nucleus (blue) in mice treated with vehicle (CTR), DNBS and DNBS+LAPE. Measurements were performed 3 days after DNBS injection. LAPE was given once a day starting from 24 h after DNBS and continued for the following days until the sacrifice. Original magnification 20×. Representative H&E-stained colon cross-sections of mice treated with vehicle, DNBS and DNBS+LAPE (300 mg/kg by oral gavage). The scale bars represented 50 μm.

**Figure 5 ijms-25-03210-f005:**
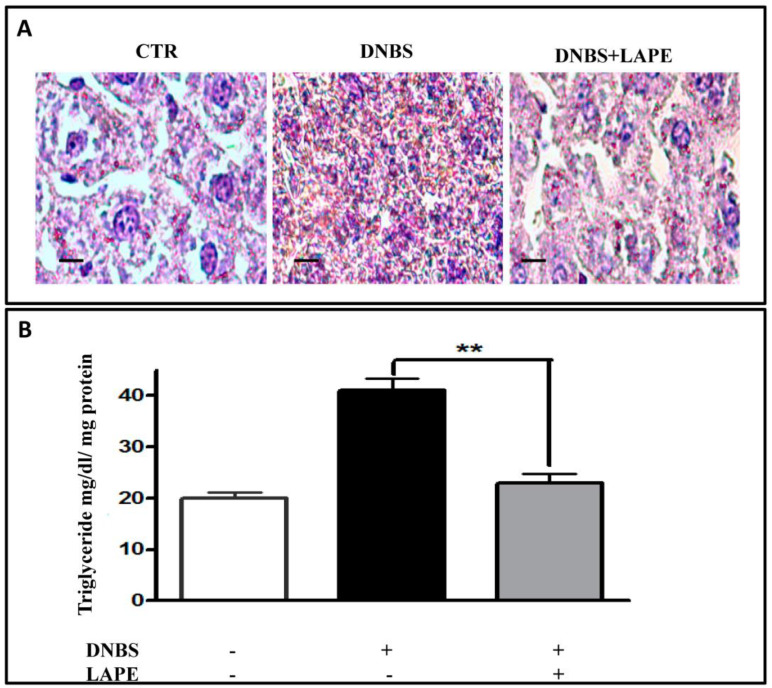
Effect of LAPE on fatty liver disease in DNBS-treated mice. Effect of LAPE (300 mg/kg, given by oral gavage) on lipid droplets (**A**) and triglycerides (TG) levels (**B**) in liver of DNBS-treated mice. Liver sections of mice treated with vehicle (CTR), DNBS or DNBS+LAPE (300 mg/kg, by oral gavage) were stained with Oil Red O. Intense red indicates lipid droplets. Measurements were performed 3 days after DNBS injection. LAPE was given once a day starting from 24 h after DNBS and continued for the following days until the sacrifice. Original magnification 20×. ** *p* < 0.01 vs. DNBS alone. Scale bars represented 100 μm.

**Figure 6 ijms-25-03210-f006:**
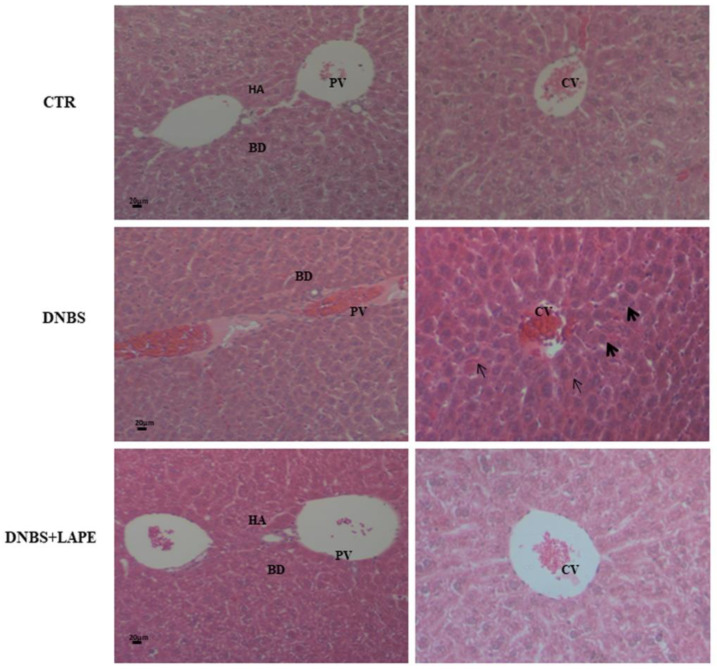
Effect of LAPE on liver damage in DNBS-treated mice. Representative H&E-stained liver cross-sections of mice treated with vehicle (CTR), DNBS and DNBS+LAPE (300 mg/kg by oral gavage). Measurements were performed 3 days after DNBS injection. LAPE was given once a day starting from 24 h after DNBS and continued for the following days until the sacrifice. Original magnification 20×. Evaluation of liver injury following arrows; thin arrows = blood sinusoids with lymphocytes infiltration; thick arrows = centrilobular hepatocytes displayed deeply stained acidophilic cytoplasm and darkly stained nuclei. BD = bile duct; CV = central vein; PV = portal vein; HA = hepatic artery. Five random fields per colon of at least four mice per group were chosen to analyze histological alterations [17].

**Figure 7 ijms-25-03210-f007:**
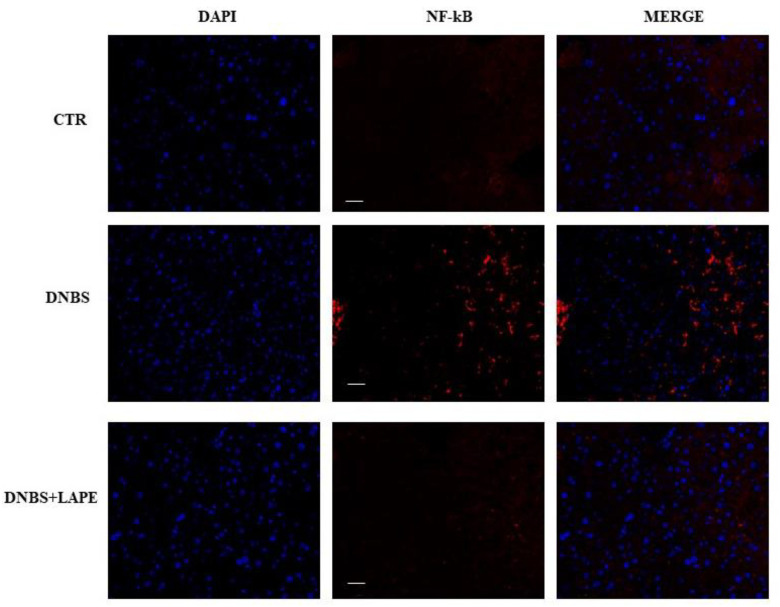
LAPE counteract NK-kB activation, in DNBS-induced colitis in mice. Immunofluorescence analysis showing the expression of NF-kB (red) of liver mice treated with vehicle (control), DNBS and DNBS plus LAPE (300 mg/kg) by oral gavage. Livers were collected three days after the induction of colitis by DNBS. Original magnification 20×. The DNBS-induced colitis in mice was treated for three consecutive days after the inflammatory insults with LAPE (300 mg/kg, by oral gavage). Scale bars represented 10 μm.

**Table 1 ijms-25-03210-t001:** Serum alanine transaminase (ALT), aspartate transaminase (AST) and TBARS levels in healthy mice (CTR) and DNBS- or DNBS+LAPE (300 mg/kg, by oral gavage)-treated mice. All results were expressed as means ± mean standard error (SEM) of at least three independent experiments. Data were statistically analyzed using a one-way ANOVA followed by Dunnett’s test.

	AST International Units per Liter (IU/L)	ALT International Units per Liter (IU/L)	TBARS μM/μg Protein
CTR	77 ± 9 *	39 ± 6 **	0.095 ± 0.01 *
DNBS	418 ± 51	185 ± 37	0.38 ± 0.09
DNBS+LAPE	263 ± 25 ***	87 ± 19	0.12 ± 0.05 *

* *p* < 0.05, ** *p* < 0.01 and *** *p* < 0.001 vs. DNBS alone.

## Data Availability

Data are contained within the article and Appendix A.

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
