# Peer review of "Polyphenol-Rich Extract from ‘Limoncella’ Apple Variety Ameliorates Dinitrobenzene Sulfonic Acid-Induced Colitis and Linked Liver Damage"

_ijms, 2024, doi:10.3390/ijms25063210_

Round 1

Reviewer 1 Report

Comments and Suggestions for Authors

This article reports on the effectiveness of a polyphenol-rich extract from a Limoncella apple variety on markers of dinitrobenzene sulfonic acid-induced colitis in male CD1 mice models. The outcomes presented will likely be of interest to readers and provide hypothesis-driving content for the scientific literature. It is especially noteworthy that the authors indicate that the extract dose in the study represented an equivalency of human consumption of 1/2 a Limoncella apple variety. The significance of the content and scientific soundness might be further improved if the following items are addressed:

Line 18-19: Appears to be an incomplete sentence, "In 30% of IBD patients the liver tests resulted increases." Consider rephrasing the statement.

Figure 1A: LAPE was shown to moderately reduce colon weight/length ratio in DNBS-treated mice, but the reduction does not appear to approach control levels. It would be helpful if the authors comment on whether the effect of LAPE on weight/length is clinically relevant.

Figure 1B: The CTR, DNBS, and DNBS+LAPE histological slices are difficult to compare.  Authors might consider pinpointing a region in each slice that can be easily compared.

Figure 2C: The IL-6 levels were reduced to levels less than control group following LAPE. The authors might consider commenting on the relevance (if any) of the significantly(?) lowered levels of IL-6.

Line 148-149:  The authors acknowledge that LAPE did not affect the level of IL-10, the main anti-inflammatory cytokine.  The scientific soundness of the report might be strengthened if the authors offered some explanation for this outcome and/or commented on the significance of this finding.

Line 187: AST and ALT are mislabeled.

Table 1: Authors might consider commenting on the ratio of AST:ALT in the control group vs. the DNBS+LAPE group.

Line 305: Authors might consider commenting on the decision to use only male mice in the study.

Line 421-422: The conclusion is overstated. This is an exploratory study in a mice model.  Further studies are needed before a conclusion can be made about the protective effects of LAPE in humans.  The authors might consider rephrasing the conclusion.

Author Response

Response to Reviewer 1

This article reports on the effectiveness of a polyphenol-rich extract from a Limoncella apple variety on markers of dinitrobenzene sulfonic acid-induced colitis in male CD1 mice models. The outcomes presented will likely be of interest to readers and provide hypothesis-driving content for the scientific literature. It is especially noteworthy that the authors indicate that the extract dose in the study represented an equivalency of human consumption of 1/2 a Limoncella apple variety. The significance of the content and scientific soundness might be further improved if the following items are addressed:

Point 1.  Line 18-19: Appears to be an incomplete sentence, "In 30% of IBD patients the liver tests resulted increases." Consider rephrasing the statement.

OUR REPLY: We have rephrased this sentence in the abstract: “In 30% of patients with IBD, blood values of liver enzymes, such as AST and ALT, were increased.”

Point 2.  Figure 1A: LAPE was shown to moderately reduce colon weight/length ratio in DNBS-treated mice, but the reduction does not appear to approach control levels. It would be helpful if the authors comment on whether the effect of LAPE on weight/length is clinically relevant.

OUR REPLY: The colon weight/colon length ratio is extensively used in literature as a macroscopic, reliable and sensitive indicator of the severity and extent of the intestinal inflammation during murine colitis. As shown in Figure 1A, the administration of the flogogen agent DNBS causes a significant and massive increase in such parameter compared to control (healthy) mice. As highlighted by reviewer, LAPE significantly reduces the colon weight/colon length, but this effect does not completely restore the levels compared to healthy mice. This result is consistent with literature, considering that DNBS develops a severe and non-reversible colitis in mice, which is rarely fully restored by anti-inflammatory agents. 

To note, the anti-inflammatory effect of LAPE is extensively confirmed by additional parameters evaluated in mice with colitis such as histological analysis (Figure 1B) and biochemical parameters (i..e, myeloperoxidase activity (Figure 2A), pro-inflammatory cytokine levels (Figure 2B,C) and intestinal permeability (Figure 2D).

Point 3.  Figure 1B: The CTR, DNBS, and DNBS+LAPE histological slices are difficult to compare.  Authors might consider pinpointing a region in each slice that can be easily compared.

OUR REPLY: we have added the pinpoint for all slice that requested.

Point 4. Figure 2C: The IL-6 levels were reduced to levels less than control group following LAPE. The authors might consider commenting on the relevance (if any) of the significantly(?) lowered levels of IL-6.

Our Reply: We thank the reviewer for highlighting this interesting point. We hypothesize that LAPE could affect IL6 levels also in healthy mice (not treated with DNBS). However, since this aspect needs further evaluation, we do not would like to speculate on the relevance of this result.

Point 5.  Line 148-149:  The authors acknowledge that LAPE did not affect the level of IL-10, the main anti-inflammatory cytokine.  The scientific soundness of the report might be strengthened if the authors offered some explanation for this outcome and/or commented on the significance of this finding.

Our Reply: In DNBS-treated mice, LAPE causes a reduction of pro-inflammatory cytokines (i.e. IL-1β and IL-6), with no effect on the anti-inflammatory cytokine IL-10. It has been demonstrated that anti-inflammatory compounds mostly exert their effect by blocking pro-inflammatory cytokine signaling rather than by boosting anti-inflammatory pathways (e.g., IL-10-mediated pathway) to re-establish intestinal homeostasis (Friedrich et al., 2019; https://doi.org/10.1016/j.immuni.2019.03.017). Our results suggest that anti-inflammatory effect of LAPE is mediated by reduction of pro-inflammatory pathways, thus we did not deepen the lack of LAPE effect on IL-10 levels. However, targeting IL-10 in intestinal pathologies remains an attractive approach and it is an increasing area of focus.

Point 6.  Line 187: AST and ALT are mislabeled.

OUR REPLY: we corrected the mistake.

Point 7.  Table 1: Authors might consider commenting on the ratio of AST:ALT in the control group vs. the DNBS+LAPE group. Although LAPE treatment of DNBS mice induced a decrease in serum enzymes AST and ALT and lipid peroxidation, demonstrating a reduction in liver damage. LAPE treatment did not reduce the AST:ALT ratio compared to the control group, this paradox could be due either to a greater decrease in serum concentrations of ALT (enzyme specific for liver damage) compared to AST (enzyme not specific only for liver damage) or to a selectivity of LAPE vs liver cells.

OUR REPLY: The elevation of AST/ALT ratio in LAPE treated DNBS-mice compared to ctr   may be due liver cells may be damaged, the mitochondria in the liver cells remain intact. As a result, only the ALT from the cytoplasm of liver cells is released into the blood, so liver function examinations indicate elevated levels of ALT and an AST/ALT ratio of <1. mitochondria in liver cells may be severely damaged As a result, AST is released from the mitochondria and cytoplasm and AST levels are evidently increased, resulting in an AST/ALT ratio of >1 (18). which indicates the degree of liver damage.

Point 8.  Line 305: Authors might consider commenting on the decision to use only male mice in the study.

Sex differences in inflammation are widely demonstrated in murine models and contribute to divergences in the severity of colitis in human. Thus, combine results for the two sexes should be done with caution because it has been observed evidence of sex-specific sensitivity to severe inflammation protocols and sex-specific differences in some of the characteristics measured (Barone et al. 2018; doi: 10.3389/fmicb.2018.00565). In our manuscript we decided to focalize the study on male response, and we did not evaluate sex-dependent differences in colitis.

Point 9.  Line 421-422: The conclusion is overstated. This is an exploratory study in a mice model.  Further studies are needed before a conclusion can be made about the protective effects of LAPE in humans.  The authors might consider rephrasing the conclusion.

OUR REPLY: We have rephrased the conclusion paragraph. “ In this study, we demonstrated that LAPE, ameliorated intestinal inflammation in mice treated with the flogogen agent DNBS. In addition, LAPE decreased serum lipid peroxidation and liver inflammation by inhibiting Nf-kB in the experimental model of colitis. However, additional experiments are needed to further validate the  results obtained in our mice model exploratory study  and elucidate the impact of  LAPE  as a protective dietary supplement in chronic intestinal inflammation leading to liver injury.”

Reviewer 2 Report

Comments and Suggestions for Authors

1. Formic acid used to prepare apple extract could influences the extract properties? Do you check the presence of formic acid trace in the lyophilized extract?

2. "The total amount of identified polyphenols in 1g of LAPE was 5.25 mg, thus according to our experimental model, each mouse receives 47,5 mg of titrated polyphenols in 9 mg of LAPE administrated daily."

If 1 g LAPE contains 5.25 mg polyphenols and each animal receives 9 mg LAPE how you have 47.5 mg plyphenols?

3. "The increase in colon permeability leads to translocation of gut microbiota to liver and to liver inflammation and damage."

Liver is affect also by different compounds that pass freely from intestine to the liver. 

4. The increasing of liver triglycerides could be induced by decreasing of protein synthesis into the liver?

5. "Since this peculiar apple variety produces small and elongated fruits with a reported average weight of 100 g, the theoretically calculated daily dose for humans could be equated with the assumption of half part of a single Limoncella apple per day."

Is difficult to say this because your extract has been administered by gastric gavage, and this is forced administration and also polyphenols are obtained by extraction and the compounds that could influence the polyphenols action are eliminated.

Author Response

Point 1.  Formic acid used to prepare apple extract could influences the extract properties? Do you check the presence of formic acid trace in the lyophilized extract?

OUR replay:  The authors are grateful for the valuable referee comments. Considering the performed extraction protocol, the methanolic extract obtained was first subjected to fractional distillation using a rotary evaporator to remove methanol. Once the methanol was completely removed, the remaining aqueous phase was diluted with water, frozen at -18°C until to obtain a stable and homogeneous solid phase, and subsequently freeze-dried for 48 h. Reasonably, after this multistep process, formic acid (boiling point 100.8°C) residues in LAPE could be negligible and insufficient to have any impact on the model used.

Point 2. "The total amount of identified polyphenols in 1g of LAPE was 5.25 mg, thus according to our experimental model, each mouse receives 47,5 mg of titrated polyphenols in 9 mg of LAPE administrated daily." If 1 g LAPE contains 5.25 mg polyphenols and each animal receives 9 mg LAPE how you have 47.5 mg plyphenols?

OUR replay: The authors are grateful for the valuable referee comments. We apologize for the typing error. In 9 mg of LAPE was contained 47.5 μg and not mg.

Point 3. "The increase in colon permeability leads to translocation of gut microbiota to liver and to liver inflammation and damage." Liver is affect also by different compounds that pass freely from intestine to the liver. 

Point 4. "Since this peculiar apple variety produces small and elongated fruits with a reported average weight of 100 g, the theoretically calculated daily dose for humans could be equated with the assumption of half part of a single Limoncella apple per day." Is difficult to say this because your extract has been administered by gastric gavage, and this is forced administration and also polyphenols are obtained by extraction and the compounds that could influence the polyphenols action are eliminated.

OUR replay: The authors are grateful for the author's comments. We are perfectly in agreement with your observations. The sentence was rearranged, as follows "Since this peculiar apple variety produces small and elongated fruits with a reported average weight of 100 g, based on LAPE polyphenolic content and its extraction rate from apple, it could be hypothesized that the calculated daily dose for humans could be equated with the assumption of half part of a single Limoncella apple per day, however, neglecting potential interactions of polyphenols in the complex apple food matrix.

Reviewer 3 Report

Comments and Suggestions for Authors

Dear Authors,

The manuscript explores the potential benefits of using apple polyphenol extract, specifically from Malus domestica cv ‘Limoncella’ (LAPE), as an intervention for hepatoxicity induced by inflammatory bowel disease (IBD). It discusses how IBD can affect various organ systems, particularly the hepatobiliary system, and increase the risk of liver damage. The study administered LAPE orally to mice with induced colitis and found significant improvements in histological bowel injury, myeloperoxidase activity, inflammatory cytokine expressions, serum lipid peroxidation, liver injury, and nuclear transcription factor-kappaB activation. These results suggest that LAPE could mitigate liver damage induced by IBD through its antioxidant and anti-inflammatory properties. Overall, the article provides valuable insights into the potential therapeutic effects of LAPE in managing hepatoxicity associated with IBD.

The manuscript under review demonstrates a commendable level of design and execution, showcasing meticulous attention to detail in both experimental methodology and selection. The authors have adeptly chosen experiments that not only address the research questions at hand but also contribute significantly to the broader investigation. The thoroughness of the study is evident in the comprehensive coverage of the investigation, leaving no aspect unexplored. From the outset, it is clear that the authors have invested considerable effort in crafting a study that is not only scientifically rigorous but also impactful in its scope.

Despite the manuscript's strengths in experimental design and execution, it is important to note the presence of grammatical and punctuational errors throughout. While the content itself is robust and well-presented, these errors may detract from the overall readability and professionalism of the manuscript. Addressing these issues through thorough proofreading and editing would enhance the clarity and coherence of the text, ensuring that the valuable insights provided by the study are effectively communicated to the reader.

Reference 10 the year is not bold.

Reference 20 – the space is needed between the journal and the year.

Reference 21, 31, 37 – please unify with the journal style

Based on the strengths observed in the manuscript's well-designed experiments and comprehensive coverage of the investigation, I recommend its publication following minor revisions.

Comments on the Quality of English Language

Despite the manuscript's strengths in experimental design and execution, it is important to note the presence of grammatical and punctuation errors throughout. MInor editing of English language required.  

Author Response

Response to Reviewer 3

The manuscript explores the potential benefits of using apple polyphenol extract, specifically from Malus domestica cv ‘Limoncella’ (LAPE), as an intervention for hepatoxicity induced by inflammatory bowel disease (IBD). It discusses how IBD can affect various organ systems, particularly the hepatobiliary system, and increase the risk of liver damage. The study administered LAPE orally to mice with induced colitis and found significant improvements in histological bowel injury, myeloperoxidase activity, inflammatory cytokine expressions, serum lipid peroxidation, liver injury, and nuclear transcription factor-kappaB activation. These results suggest that LAPE could mitigate liver damage induced by IBD through its antioxidant and anti-inflammatory properties. Overall, the article provides valuable insights into the potential therapeutic effects of LAPE in managing hepatoxicity associated with IBD.

The manuscript under review demonstrates a commendable level of design and execution, showcasing meticulous attention to detail in both experimental methodology and selection. The authors have adeptly chosen experiments that not only address the research questions at hand but also contribute significantly to the broader investigation. The thoroughness of the study is evident in the comprehensive coverage of the investigation, leaving no aspect unexplored. From the outset, it is clear that the authors have invested considerable effort in crafting a study that is not only scientifically rigorous but also impactful in its scope.

Despite the manuscript's strengths in experimental design and execution, it is important to note the presence of grammatical and punctuational errors throughout. While the content itself is robust and well-presented, these errors may detract from the overall readability and professionalism of the manuscript. Addressing these issues through thorough proofreading and editing would enhance the clarity and coherence of the text, ensuring that the valuable insights provided by the study are effectively communicated to the reader.

Reference 10 the year is not bold.

Reference 20 – the space is needed between the journal and the year.

Reference 21, 31, 37 – please unify with the journal style

OUR REPLY: We have corrected the mistakes on the reference.